# *Glochidion littorale* Leaf Extract Exhibits Neuroprotective Effects in *Caenorhabditis elegans* via DAF-16 Activation

**DOI:** 10.3390/molecules26133958

**Published:** 2021-06-28

**Authors:** Abdel Fawaz Bagoudou, Yifeng Zheng, Masahiro Nakabayashi, Saroat Rawdkuen, Hyun-Young Park, Dhiraj A. Vattem, Kenji Sato, Soichiro Nakamura, Shigeru Katayama

**Affiliations:** 1Graduate School of Medicine, Science and Technology, Shinshu University, 8304 Minamiminowa, Kamiina, Nagano 399-4598, Japan; 18hs552b@shinshu-u.ac.jp (A.F.B.); snakamu@shinshu-u.ac.jp (S.N.); 2Institute for Biomedical Sciences, Shinshu University, 8304 Minamiminowa, Kamiina, Nagano 399-4598, Japan; zhengyf@shinshu-u.ac.jp (Y.Z.); 15aa408a@shinshu-u.ac.jp (M.N.); 3School of Agro-Industry, Mae Fah Luang University, 333 Moo 1, Thasud, Muang, Chiang Rai 57100, Thailand; saroat@mfu.ac.th; 4Edison Biotechnology Institute, Konneker Research Laboratories, Ohio University, Athens, OH 45701, USA; parkh4@ohio.edu (H.-Y.P.); vattem@ohio.edu (D.A.V.); 5College of Health Sciences & Professions, Ohio University, Athens, OH 45701, USA; 6Graduate School of Agriculture, Kyoto University, Kyoto 606-8502, Japan; kensato@kais.kyoto-u.ac.jp

**Keywords:** *Caenorhabditis elegans*, leaf extract, neuroprotection, antioxidant activity, DAF-16

## Abstract

A number of plants used in folk medicine in Thailand and Eastern Asia are attracting interest due to the high bioactivities of their extracts. The aim of this study was to screen the edible leaf extracts of 20 plants found in Thailand and investigate the potential neuroprotective effects of the most bioactive sample. The total phenol and flavonoid content and 2,2-diphenyl-1-picrylhydrazyl radical-scavenging activity were determined for all 20 leaf extracts. Based on these assays, *Glochidion littorale* leaf extract (GLE), which showed a high value in all tested parameters, was used in further experiments to evaluate its effects on neurodegeneration in *Caenorhabditis elegans*. GLE treatment ameliorated H_2_O_2_-induced oxidative stress by attenuating the accumulation of reactive oxygen species and protected the worms against 1-methyl-4-phenylpyridinium-induced neurodegeneration. The neuroprotective effects observed may be associated with the activation of the transcription factor DAF-16. The characterization of this extract by LC-MS identified several phenolic compounds, including myricetin, coumestrin, chlorogenic acid, and hesperidin, which may play a key role in neuroprotection. This study reports the novel neuroprotective activity of GLE, which may be used to develop treatments for neurodegenerative diseases such as Parkinson’s syndrome.

## 1. Introduction

Neurodegenerative disorders including Alzheimer’s disease and Parkinson’s disease (PD) pose major health and financial concerns to global health care organizations [1]. Although the human lifespan has increased in the last few decades in industrialized countries, the prevalence of age-related diseases has also increased. The incidence of late-onset disorders such as neurological disruptions is expected to increase rapidly over the next few decades. Therefore, it is crucial to encourage studies and perform clinical trials on compounds that may have the potential to cure, prevent, or at least delay the onset of neurodegenerative diseases [2]. One of the characteristic features of PD is the progressive loss of dopaminergic (DA) neurons in the substantia nigra [3]. In PD pathogenesis, increased production of reactive oxygen species (ROS) plays a key role in the loss of DA cells [4]. Therefore, the reduction in oxidative stress is considered a promising therapeutic approach in PD treatment [5]. The 1-methyl-4-phenylpyridinium (MPP^+^), which inhibits mitochondrial complex I activity, can induce PD-like symptoms in humans and animal models [6].

The use of *Caenorhabditis elegans* as an in vivo model provides certain advantages in the study of PD [7]. The nematode is simple, inexpensive, and has a short life cycle. It supports studies involving large-scale analyses. Moreover, the neuronal network of *C. elegans* has been mapped completely. It contains 8 DA neurons and PD-related homologous genes [8]. Neurodegeneration, which mimics parkinsonian symptoms, can be induced in *C. elegans* via treatment with neurotoxins such as MPP^+^ [9].

Natural antioxidant compounds represent attractive sources for developing drugs to treat neurodegenerative diseases due to their neuroprotective effects in animal models and low toxicity [3]. Polyphenols are known to be among the most abundant antioxidants in the human diet [10]. It has also been established that oxidative processes are involved in many pathologies, including neurodegeneration, cancer, diabetes, cardiovascular and anti-inflammatory diseases. Hence, finding polyphenols exhibiting antioxidant properties from natural sources could contribute toward preventing or treating those pathologies. This study focused on extracts from the edible leaves of plants found in Thailand. Most varieties cultivated widely in northern and southern Thailand have been used as folk medicine against general injuries and diseases; however, there are few reports concerning their neuroprotective effects.

In this study, we first screened the extracts of edible leaves from 20 plants cultivated in Thailand and assessed their phenolic and flavonoid contents and their 2,2-diphenyl-1-picrylhydrazyl (DPPH) radical-scavenging activity. The effects of *Glochidion littorale* leaf extract (GLE), which showed a high value in all tested parameters, were evaluated on *C. elegans* with neurodegeneration. Furthermore, the potential pathways involved in the neuroprotective effect of GLE were examined, along with the identification of the main components in GLE.

## 2. Results

### 2.1. Screening of Thai Plant Leaves

Crude extracts of edible leaves from plants cultivated in Thailand were prepared by ultrasonication. The leaf extracts of 20 plants were screened for their phenolic and flavonoid contents and antioxidant activity by DPPH radical-scavenging assay. Few of the tested samples, such as *Glochidion sphaerogynum* and *Mentha piperita,* were found to possess high radical-scavenging activity with low phenolic and flavonoid content, whereas certain samples, such as *Clinacanthus nutans* and *Ocimum* × *citriodorum*, exhibited the opposite trend (Table 1). The leaf extract of *G. littorale* showed high DPPH radical-scavenging activity as well as high phenolic and flavonoid content. Therefore, the bioactivities associated with *G. littorale* were further investigated.

### 2.2. GLE Enhanced Resistance against Oxidative Stress via DAF-16 in C. Elegans

The effect of GLE on the survival of N2 worms under oxidative stress was investigated. Treatment with H_2_O_2_ (5 mM) induced 75% death in the control group, whereas co-treatment with 50 µg/mL and higher concentrations of GLE was associated with a high survival rate (Figure 1A). Among the tested concentrations of GLE, 100 µg/mL and 200 µg/mL were associated with the highest survival rates (82.0% and 88.2%, respectively). Therefore, these two concentrations were used in subsequent experiments. To evaluate the antioxidant effect of GLE in vivo, the intracellular ROS levels were measured in wild-type nematodes using 2’,7’-dichlorodihydrofluorescein diacetate (H_2_DCF-DA), a well-known fluorescence probe for detecting intracellular ROS production. Significant decreases in the fluorescence intensities in the GLE-treated groups were observed compared to that in the untreated group (Figure 1B), confirming the antioxidant property of GLE.

As the transcription factor DAF-16 is known to play a key role in regulating oxidative stress [11], it was hypothesized that GLE may target DAF-16. The *C. elegans* strain CF1038, which is a DAF-16 loss-of-function mutant strain, was used to determine the survival rate of worms treated with and without GLE. In H_2_O_2_-induced oxidative stress, GLE treatment did not increase the survival rate of transgenic worms (Figure 1C).

### 2.3. GLE Treatment Reduced the Lethality of MPP^+^-Induced DA Neurotoxicity via DAF-16 in C. Elegans

*C. elegans* possesses 8 DA neurons [8]. Selective degeneration of these DA neurons was evaluated after exposure to MPP^+^. The treatment of wild-type N2 worms with 0.75 mM MPP^+^ resulted in a remarkable decrease in survival (Figure 2). However, co-treatment with GLE significantly increased the survival of the worms. The effect of GLE treatment on *daf-16* mutant worms was investigated. As shown in Figure 3 and Table 2, GLE treatment did not increase the survival of these worms after exposure to MPP^+^ compared to that in the control group. These results suggest that DAF-16 may be required for mediating the neuroprotective effect of GLE in *C. elegans*. Next, a DAF-2 loss-of-function mutant strain, CB1370, was used to determine whether DAF-2 was involved in the observed neuroprotective effects. As shown in Figure 4 and Table 3, the median and maximum survival significantly increased in *daf-2* mutant worms treated with GLE.

### 2.4. Effects of GLE on DAF-16 Localization

It has been demonstrated that DAF-16 activation is regulated by its nuclear accumulation [12]. Subsequently, we investigated whether GLE could induce the nuclear accumulation of DAF-16 in a transgenic strain TJ356 that expresses a DAF-16::GFP fusion protein. Results showed that after 48 h of incubation with 100 µg/mL GLE, the green fluorescence intensity of DAF-16 in the nucleus increased significantly compared to that in the untreated group (Figure 5).

### 2.5. Phytochemical Characterization in GLE

LC-MS was conducted for profiling the phytochemicals in GLE, and its results are presented in Figure 6. The chromatographic peaks were identified by comparing the MS data with databases based on the search of m/z values of molecular ion peaks in the positive mode [M + H]^+^. Consequently, myricetin, coumestrin, chlorogenic acid, and hesperidin were detected as the major compounds (Table 4).

## 3. Discussion

Plant extracts are a rich source of natural bioactive compounds. Many studies have evaluated plant extracts used in Southeast Asian countries, including Thailand, where these extracts are components of folk medicine [13,14]. In this study, the extracts of 20 edible plant leaves from Thailand were screened, and *G. littorale* was selected for further investigation because it showed high phenol content, flavonoid content, and radical-scavenging activity. Several studies have investigated various species of the genus *Glochidion* [15,16,17,18,19]; however, there are few studies concerning the functional properties and constituents of *G. littorale.* Our data showed that GLE protected *C. elegans* against H_2_O_2_-induced oxidative stress by reducing intracellular ROS accumulation. This might have been due to the high content of phenolic compounds such as flavonoids, which are known to possess strong antioxidant activity [20]. These findings are similar to those obtained by Duangjan et al. (2019), who showed that *G. zeylanicum* leaf extracts can protect *C. elegans* against oxidative stress [21]. The insulin/insulin-like signaling (IIS) pathway regulates growth, stress responsiveness, and longevity in *C. elegans* [22,23]. We found that *daf-16* null mutant *C. elegans* treated with GLE were susceptible to oxidative stress. This result suggests that the antioxidant effect of GLE in reducing oxidative stress in nematodes is possibly involved in not only radical-scavenging activity but also the regulation of the DAF-16 transcription factor.

The protective effects of GLE against MPP^+^-induced toxicity in *C. elegans* were examined. DA neurons in nematodes take up MPP^+^ mainly via high-affinity DA transporters, which is similarly observed in mammals. The accumulation of MPP^+^ inside the neurons inactivates the mitochondrial complex I of the respiratory chain and induces cell death [24,25,26,27]. GLE treatment was found to significantly reduce the lethality associated with MPP^+^ treatment in wild-type worms. The IIS pathway is modulated by insulin-like peptides through the DAF-2 receptor in *C. elegans* [28]. Under normal conditions, the IIS pathway inhibits the phosphorylation of DAF-16 and prevents its nuclear translocation. In *daf-2* null mutants, the GLE-treated group survived longer than the control group. In contrast, no difference in survival was observed between the control group and the GLE-treated group containing *daf-16* null mutant worms. It is known that downregulated DAF-2 signaling facilitates the entry of DAF-16 into the nucleus, where it can upregulate the expression of target genes and control stress resistance and longevity [29]. This may explain why *daf-2* mutant worms treated with GLE showed a relatively higher survival. Furthermore, an increased nuclear accumulation of DAF-16 in worms treated with GLE was observed using transgenic TJ356 DAF-16::GFP*C. elegans*. Cumulatively, these results indicated that GLE might have exhibited its neuroprotective effects via activation of DAF-16.

LC-MS profiling led to the identification of 11 phytochemical compounds in GLE. Myricetin identified in the main peak of GLE is a flavonoid widely found in many plants and is well known to exhibit protective effects against oxidative stress. A previous study has demonstrated that myricetin extended the lifespan of *C. elegans* by diminishing stress-induced ROS accumulation and the pro-longevity effects of myricetin were dependent on DAF-16 [30]. Chlorogenic acid has also been reported to exhibit pro-longevity effects via the attenuation of oxidative stress in *C. elegans* [31]. Considering these findings, the neuroprotective effects of GLE were mainly induced by flavonoids such as myricetin, and GLE might be a suitable candidate for the management of neurodegenerative diseases.

In conclusion, our study demonstrated that GLE possessed strong antioxidant activity, which reduced oxidative stress in *C. elegans*. The extract also showed neuroprotective activity against MPP^+^-induced neurotoxicity in *C. elegans*. Various experiments performed using different transgenic worms suggested the possible involvement of the DAF-16 transcription factor in the observed neuroprotection. The high content of phenolic compounds, including flavonoids present in GLE, may be responsible for the observed stress resistance and neuroprotective properties. Further studies should identify the target genes involved in the neuroprotection mechanism.

## 4. Materials and Methods

### 4.1. Materials

The leaves of 20 different plants (Table 1) were obtained from a local market in Chiang Rai, Thailand. All reagents were of analytical grade. DPPH and H_2_DCF-DA were obtained from Sigma-Aldrich (St. Louis, MO, USA). All other chemicals were obtained from Wako Pure Chemical Industries (Osaka, Japan).

### 4.2. Preparation of Leaf Extracts

Leaf samples were frozen in liquid nitrogen, and then powdered samples (5 g) were mixed in 100 mL of distilled water at 45 °C for 30 min, following sonication using a Branson SLPe Sonifier (Branson, North Billerica, MA, USA) at 35 kHz. The extract was filtered and freeze-dried to obtain a powdered sample.

### 4.3. Total Phenolic Contents

The Folin-Ciocalteu method was used to determine the total phenolic content. Briefly, 11.4 μL of the extract (1 mg/mL) was mixed with 227.3 μL of 2% (*w*/*v*) Na_2_CO_3_ solution, and then the mixture was allowed to stand at room temperature for 2 min. After addition of 11.4 μL of 10% (*v*/*v*) Folin-Ciocalteu reagent. The incubation in the dark was conducted for 30 min. Subsequently, the absorbance was measured at 750 nm using a microplate reader (Nivo 3F Multimode Plate Reader, PerkinElmer, Waltham, MA, USA). Gallic acid was used as a standard for the calibration curve. The total phenolic content was expressed as gallic acid equivalents (mg gallic acid equivalent/g of plant extract).

### 4.4. Total Flavonoid Contents

The aluminum chloride colorimetric method was used to measure the total flavonoid content. Briefly, 25 μL of the extract (2 mg/mL) was mixed with 7.5 μL of 5% (*w*/*v*) NaNO_2_ solution and 152.5 μL of distilled water. After 6 min, 15 μL of 10% (*w*/*v*) AlCl_3_ solution was added and allowed to stand for 5 min. Then, 50 μL of 1 M NaOH solution was added to the mixture. Subsequently, the mixture was incubated in the dark for 15 min, and the absorbance was measured at 510 nm using a microplate reader. The total flavonoid content was calculated by generating a calibration curve using quercetin as a standard, and the results were expressed as quercetin equivalent (mg quercetin equivalent/g of plant extract).

### 4.5. Free Radical-Scavenging Activity

The capacity to scavenge free radicals was assessed using DPPH assays [32]. Briefly, 100 μL of the extract (1 mg/mL) were mixed with 100 μL of DPPH solution. After 30 min, the absorbance was measured at 517 nm using a microplate reader. The results were expressed as a percentage of inhibition of the DPPH radicals.

### 4.6. C. Elegans Maintenance

Wild-type N2, CF1038 (*daf-16(mu86) I*), CB1370 (*daf-2(e1370) III*), and TJ356 (*zIs356 [daf-16p::daf-16a/b::GFP + rol-6(su1006)]*) strains and their diet, *Escherichia coli* OP50, were obtained from the Caenorhabditis Genetics Center (Minneapolis, MN, USA). According to the standard protocols, N2, CF1038, and TJ356 strains and CF 1370 strain were maintained at 20 and 15 °C, respectively, on nematode growth medium (NGM) agar plates containing heat inactivated *E. coli* OP50 [33]. S-complete solution was prepared according to previously described literature [34].

### 4.7. Oxidative Stress Assays

Oxidative stress was induced by treating wild-type (N2) and *daf-16* mutant (CF1038) worms with H_2_O_2_. L1 larvae were added to 96-well plates at an average of 15 nematodes per well in a 40 μL solution containing *E. coli* OP50. Five mM of H_2_O_2_ solution and the tested GLE dissolved in S-complete solution were added to achieve a final volume of 50 μL per well. L1 larvae were incubated for 48 h with H_2_O_2_ alone or in the presence of various concentrations of GLE, and worm viability was visually inspected under a stereomicroscope. The results from the H_2_O_2_-treated groups were normalized and expressed as a percentage of normal controls. The results were obtained from three independent experiments (100–160 worms/treatment in each experiment).

### 4.8. Intracellular ROS Levels

Intracellular ROS levels were determined using the H_2_DCF-DA probe. L1 larvae of wild-type N2 worms were treated with 5 mM H_2_O_2_ and GLE at different concentrations in S-solution for 48 h in black 96-well plates; each well comprised a minimum of 100 worms. Worms were subsequently incubated with 25 μM H_2_DCF-DA in the dark at 20 °C for 1 h. After incubation, the fluorescence intensity was measured at wavelengths of 485/530 nm using a Powerscan HT microplate reader (DS Pharma Biomedical, Osaka, Japan).

### 4.9. Neurotoxicity Assay

Neurotoxicity was induced by treating wild-type (N2) and transgenic (CF1038 and CB1370) worms with MPP^+^. L1 larvae were added to 96-well plates (15 worms/well) in a 40 μL solution containing *E. coli* OP50. Worms were then incubated with 50 μL of 0.75 mM MPP^+^ alone or in the presence of different concentrations of the tested sample for 48 h. After incubation, worm viability was visually inspected under a stereomicroscope. Live worms were counted every 12 h post treatment until no live worms remained. The results of the MPP^+^-treated groups were normalized and expressed as a percentage of the normal controls. The results were obtained from three independent experiments (80–130 worms/treatment in each experiment).

### 4.10. Nuclear Localization of DAF-16

Transgenic *C. elegans* TJ356, which expresses a DAF-16-GFP fusion protein, was used to examine the intracellular distribution of DAF-16. L1 stage nematodes were treated with GLE for 48 h at 20 °C. The worms were then transferred to a 2% agarose pad on a glass slide and anesthetized by adding one drop (approximately 20 μL) of 25 μM sodium azide to the agarose pad. The expression of GFP was examined via fluorescence microscopy (EVOS fl; Advanced Microscopy Group, Bothell, WA, USA). The mean fluorescence intensity of DAF-16 in the nuclei was analyzed using Image J software (National Institutes of Health, Bethesda, MD, USA).

### 4.11. Phytochemical Profiling Using LC-MS

The leaf extract was analyzed using the LCMS-8040 (Shimadzu). Mass spectra were acquired over a range of m/z 50–1000 using the Q3 scan mode. The solution was injected onto an Inertsil ODS-3 (250 × 2.1 mm, 5 µm, GL Sciences, Tokyo Japan) at a column temperature at 40 °C using a gradient of (A) 0.1% formic acid and (B) acetonitrile/water (80/20) containing 0.1% formic acid. The following gradient with a flow rate of 0.2 mL/min was used: 0–100% B (0–45 min), 100% B (45–50 min), and 0% B (50–60 min). Compounds were putatively identified by matching the experimental m/z values to the library of theoretical calculated m/z values in databases, including the Human Metabolome Database and the METLIN database.

### 4.12. Statistical Analysis

Data were expressed as the mean ± standard deviation for each group. The significant difference between the two groups was assessed using the *t*-test, whereas the difference between three and more groups was assessed using one-way ANOVA, followed by Tukey’s post-hoc comparison test. Statistical significance was set at *p* < 0.001 and *p* < 0.0001. For lifespan assays, C. elegans survival was plotted using Kaplan–Meier survival curves and analyzed via log-rank tests using GraphPad Prism software (version 9.01; GraphPad Software, San Diego, CA, USA).

## Figures and Tables

**Figure 1 molecules-26-03958-f001:**
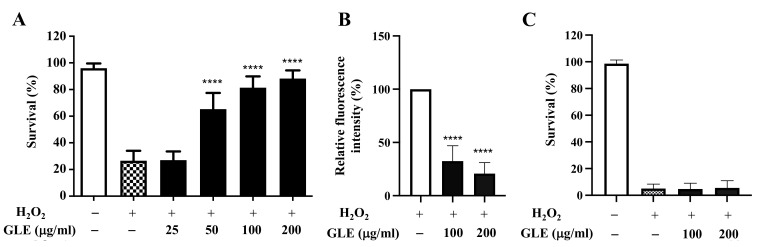
Effect of *Glochidion litorale* leaf extract (GLE) on stress resistance in wild-type and *daf-16* mutant *Caenorhabditis elegans*. (**A**) Effect of GLE against H_2_O_2_-induced toxicity in wild-type worms. (**B**) Intracellular reactive oxygen species (ROS) contents in wild-type worms. (**C**) Effect of GLE against H_2_O_2_-induced toxicity in *daf-16* mutant worms. Experiments were performed in triplicate. Data are presented as mean ± standard deviation (SD). **** *p* < 0.0001 compared to H_2_O_2_-treated worms.

**Figure 2 molecules-26-03958-f002:**
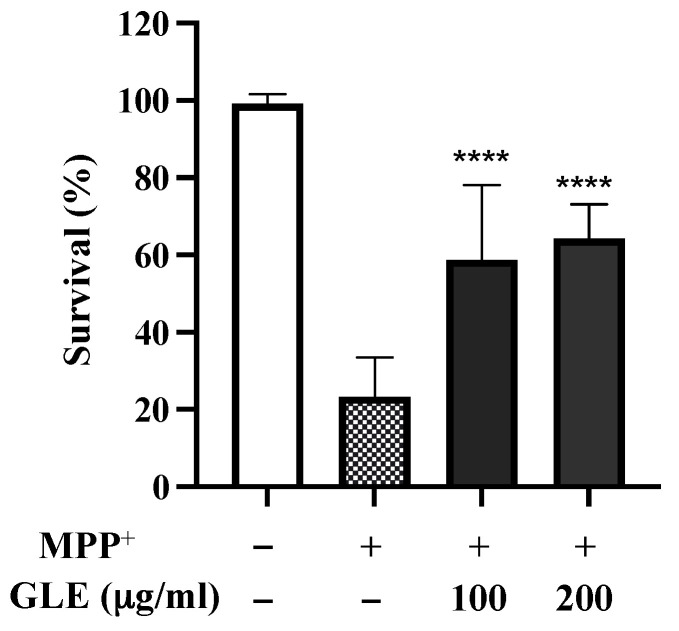
Effect of GLE on 1-methyl-4-phenylpyridinium ion (MPP^+^)-induced neurotoxicity in N2 *C. elegans.* The effects of GLE (100 and 200 µg/mL) on MPP^+^-induced toxicity were evaluated. Experiments were performed in triplicate. Data are presented as mean ± SD. **** *p* < 0.0001 compared to MPP^+^-treated worms.

**Figure 3 molecules-26-03958-f003:**
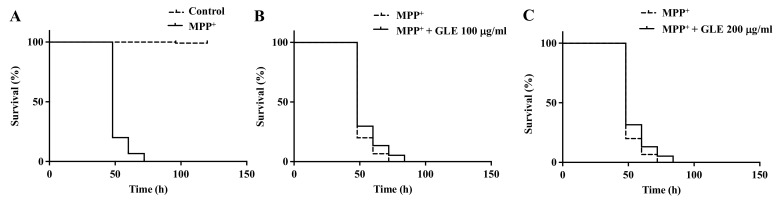
Effect of GLE on MPP^+^-induced neurotoxicity in *daf-16* mutant *C. elegans*. (**A**) Lifespan curve of worms in the presence or absence of MPP^+^. (**B**) Lifespan curve of worms with MPP^+^-induced toxicity treated with 100 µg/mL GLE. (**C**) Lifespan curve of worms with MPP^+^-induced toxicity treated with 200 µg/mL GLE. Each experiment was repeated independently at least thrice, and one of the representative data is shown.

**Figure 4 molecules-26-03958-f004:**
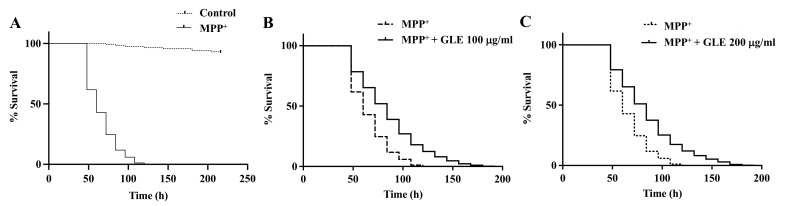
Effect of GLE on MPP^+^-induced neurotoxicity in *daf-2* mutant *C. elegans*. (**A**) Lifespan curve of worms in the presence or absence of MPP^+^. (**B**) Lifespan curve of worms with MPP^+^-induced toxicity treated with 100 µg/mL GLE. (**C**) Lifespan curve of worms with MPP^+^-induced toxicity treated with 200 µg/mL GLE. Each experiment was repeated independently at least thrice, and one of the representative data is shown.

**Figure 5 molecules-26-03958-f005:**
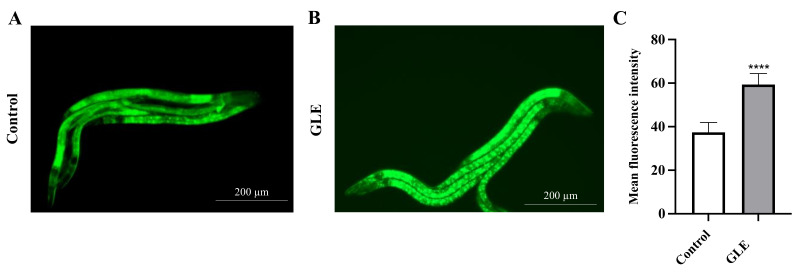
Effect of GLE on DAF-16 localization. (**A**) Untreated worms. (**B**) Worms treated with 100 µg/mL GLE. (**C**) Quantification of DAF-16::GFP nuclear accumulation in GLE and GLE-free conditions. The scale bar shows 200 µm. Each experiment was repeated independently at least thrice. Significant differences were analyzed using the *t*-test method; **** *p* < 0.0001 as compared with control.

**Figure 6 molecules-26-03958-f006:**
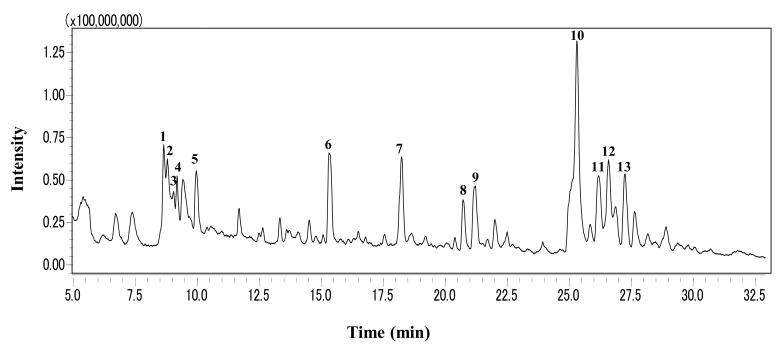
LC-MS profile of GLE. The total ion chromatogram was obtained by a triple quadrupole mass spectrometer operated in the positive electrospray ionization mode.

**Table 1 molecules-26-03958-t001:** Properties of the plants investigated in this study.

No.	Scientific Name	Phenolic Content (mg GAE ^1^/g)	Flavonoid Content (mg quercetin/g)	DPPH ^2^ Radical-Scavenging Activity (%)
1	*Clinacanthus nutans*	3.306	1.799	41.058
2	*Gymnema inodorum*	1.183	0.959	33.890
3	*Glochidion sphaerogynum*	0.709	0.781	51.705
4	*Anethum graveolens*	0.825	0.387	29.367
5	*Spilanthes acmella*	0.552	0.243	18.580
6	*Acacia pennata*	5.031	1.563	51.914
7	*Mentha piperita*	1.107	0.761	59.151
8	*Glochidion littorale*	20.104	4.527	78.984
9	*Ocimum sanctum* Linn.	0.131	0.076	45.442
10	*Ocimum basilicum* Linn.	1.342	0.771	51.635
11	*Ocimum × citriodorum*	2.446	1.612	39.666
12	*Azadirachta indica*	13.744	2.725	79.819
13	*Morus Alba*	6.190	4.019	54.488
14	*Moringa oleifera*	1.696	5.696	18.928
15	*Psidium guajava* Linn.	3.414	2.937	61.865
16	*Melientha suavis* Pierre	2.263	2.433	40.362
17	*Pandanus amaryllifolius*	1.409	1.050	36.395
18	*Zanthoxylum limonella*	3.128	1.469	46.555
19	*Piper sarmentosum*	0.584	0.487	24.217
20	*Citrus maxima*	11.690	2.461	79.193

^1^ GAE, gallic acid equivalent; ^2^ DPPH, 2,2-diphenyl-1-picrylhydrazyl.

**Table 2 molecules-26-03958-t002:** Survival of *daf-16* mutant *C. elegans* treated with MPP^+^.

Survival Time	MPP^+ 1^	MPP^+^ + GLE (100 µg/mL)	MPP^+^ + GLE (200 µg/mL)
Median (h)	48.0 ± 1.2	48.0 ± 1.7	48.0 ± 1.6
Maximum (h)	72.0 ± 1.5	72.0 ± 2.1	72.0 ± 1.8

^1^ MPP^+^, 1-methyl-4-phenylpyridinium.

**Table 3 molecules-26-03958-t003:** Survival of *daf-2* mutant *C. elegans* treated with MPP^+^.

Survival Time	MPP^+ 1^	MPP^+^ + GLE (100 µg/mL)	MPP^+^ + GLE (200 µg/mL)
Median (h)	60.0 ± 2.6	84.0 ± 3.9 ^***^	84.0 ± 4.0 ^***^
Maximum (h)	108.0 ± 5.3	180.0 ± 5.5 ^****^	192.0 ± 9.6 ^****^

*** *p* < 0.001, **** *p* < 0.0001 vs. ^1^ MPP^+^-treated worms.

**Table 4 molecules-26-03958-t004:** Compounds identified from the chromatogram of GLE.

Peak	Retention Time (min)	[M + H]^+^ (*m*/*z*)	Identified Compounds	Theoretical Mass	Mass Error (ppm)
1	8.7	431.0973	Coumestrin	430.0900	6
2	8.8	299.2005	All-trans-3,4-didehydro-retinoic acid	298.1933	1
3	9.0	248.2009	Lycopodine	247.1936	3
4	9.2	289.0707	2-Hydroxynaringenin	288.0634	10
5	10.0	166.1226	Hordenine	165.1154	1
6	15.3	355.1024	Chlorogenic acid	354.0952	7
7	18.3	611.1970	Hesperidin	610.1898	5
8	20.7	449.1078	Quercitrin	448.1006	17
9	21.2	459.0922	Epigallocatechin gallate	458.0849	3
10	25.3	319.0448	Myricetin	318.2370	5
11	26.2	465.1028	Isoquercitrin	464.0955	3
12	26.6	465.3575	Unknown	-	-
13	27.3	567.3038	Unknown	-	-

## Data Availability

The data are available by the corresponding author upon reasonable request.

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
