# Peer review of "Glochidion littorale Leaf Extract Exhibits Neuroprotective Effects in Caenorhabditis elegans via DAF-16 Activation"

_molecules, 2021, doi:10.3390/molecules26133958_

Round 1

Reviewer 1 Report

I analyzed the manuscript. There is some things, as you can see in attachment. I hope that the mistakes are errors in text editing. I also make my own opinion on some theoretical aspects.

1) In paragraph 2.2. GLE enhanced resistance against oxidative stress via DAF-16 in C. elegans

at row 90, the authors write:

“μg/mL were associated with the highest survival rates (82% and 88.2%, respectively).”

I think that is better to use the same precision degree: 82.0% and 88.2% instead of 82% and 88.2%

2) In paragraph 2.3. GLE treatment reduced the lethality of MPP+-induced DA neurotoxicity via DAF-16 in C. 108 elegans

at row 121, Figure 2. the authors write:

MPP+                +          +          +          +

GLE (mg/ml)   –          –          100      200

At the control group (first column), the MPP+ must be negative. I suggest to make the change:

MPP+                –          +          +          +

GLE (mg/ml)   –          –          100      200

3) In paragraph 4.3. Total phenolic contents

at rows 231 – 232 the authors write:

phenol reagent. After 20 min, the mixture was neutralized by the addition of 50 μL of

231

Na2CO3 solution (7.5%; v/v), and the incubation in the dark was conducted for 20 min.

232

Sodium carbonate is a solid substance, so the mode in which the concentration of a Na2CO3 solution is presented is wrong, because it is not measured a volume of Na2CO3. I suggest to change with “50 μL of Na2CO3 solution (7.5%; m/v) because the correct way to prepare this solution is to measure 7.5 grams of Na2CO3 and to dissolve and dilute at 100 mL with water, and in this case the concentration is expressed as mass / weight percent.

4) In paragraph 4.4. Total flavonoid contents

at rows 238 - 240 the authors write:

content. Briefly, 50 μL of the extract (100 μg/mL) was mixed with 150 μL of 95% ethanol,

238

10 μL of 10% (v/v) AlCl3 solution, and 10 μL of 1 M NaOAc solution. Subsequently, the

239

mixture was incubated in the dark for 40 min and the absorbance was measured at 415 nm.

240

  1. Aluminium chloride is a solid substance, so the mode in which the concentration of a AlCl3 solution is presented is wrong, because it is not measured a volume of AlCl3. I suggest to change with “10 μL of 10% (m/v) AlCl3 solution because the correct way to prepare this solution is to measure 10 grams of Al2Cl3 and to dissolve and dilute at 100 mL with water, and in this case the concentration is expressed as mass / weight percent.

  1. What is NaOAc solution? From my experience in liquid chromatography I suppose that is sodium acetate, but I am not sure. So, I suggest to write the chemical formula or the name of the substance in bracket.

5) In paragraph 4.11. Phytochemical profiling using LC-MS

at rows 296 - 297 the authors write:

temperature at 40°C using a gradient of (A) 0.1% formic acid and (B) 80% acetonitrile con-

296

taining 0.1% formic acid. The following gradient with a flow rate of 0.2 mL/min was used:

297

I understand the composition of the used mobile phase, I suppose that 80% acetonitrile means a mixture of acetonitrile / water in 80 / 20 ratio. If I am right, I think that is better to write: “using a gradient of (A) 0.1% formic acid and (B) acetonitrile / water (80/20) containing 0.1% formic acid.”

6) In paragraph 4.11. Phytochemical profiling using LC-MS

at rows 298 - 299 the authors write:

0−100% B (0−45 min), 100% B (45−50 min), and 0% B (50−60 min). Compounds were puta-

296

tively identified by matching the m/z and fragmentation patterns in mass spectral data

297

I suggest to write “Compounds were calitatively identified by matching the m/z and fragmentation patterns in mass spectral data-bases”

or, simply:

“Compounds were identified by matching the m/z and fragmentation patterns in mass spectral data-bases”

Author Response

We appreciate the time and effort invested by the reviewers and the editor. According to reviewer’s comments, we have carefully revised the manuscript and now believe its quality to be substantially improved. Point-by-point responses to each comment can be found below.

Comments and Suggestions for Authors

I analyzed the manuscript. There is some things, as you can see in attachment. I hope that the mistakes are errors in text editing. I also make my own opinion on some theoretical aspects.

1) In paragraph 2.2. GLE enhanced resistance against oxidative stress via DAF-16 in C. elegans

at row 90, the authors write:

“μg/mL were associated with the highest survival rates (82% and 88.2%, respectively).”

I think that is better to use the same precision degree: 82.0% and 88.2% instead of 82% and 88.2%

Response: Thank you for pointing this out. We have revised “82%” to “82.0%” (Revised page 3, line 95).

2) In paragraph 2.3. GLE treatment reduced the lethality of MPP+-induced DA neurotoxicity via DAF-16 in C. 108 elegans

at row 121, Figure 2. the authors write:

MPP               +         +          +           +        +

GLE (mg/ml)   –          –          100      200

At the control group (first column), the MPP+ must be negative. I suggest to make the change:

MPP+            –          +          +         +

GLE (mg/ml)   –          –          100      200

Response: Thank you for pointing this out. We have revised “+” to “–”/ in the column of Figure 2.

3) In paragraph 4.3. Total phenolic contents

at rows 231 – 232 the authors write:

phenol reagent. After 20 min, the mixture was neutralized by the addition of 50 μL of       231

Na2CO3 solution (7.5%; v/v), and the incubation in the dark was conducted for 20 min.     232

Sodium carbonate is a solid substance, so the mode in which the concentration of a Na2CO3 solution is presented is wrong, because it is not measured a volume of Na2CO3. I suggest to change with “50 μL of Na2CO3 solution (7.5%; m/v) because the correct way to prepare this solution is to measure 7.5 grams of Na2CO3 and to dissolve and dilute at 100 mL with water, and in this case the concentration is expressed as mass / weight percent.

Response: The phase “Na2CO3 solution (7.5%; v/v)” and the concentration of this solution were incorrect. We have revised it to “5% (w/v) NaNO2 solution” (Revised page 9, line 250).

4) In paragraph 4.4. Total flavonoid contents

at rows 238 - 240 the authors write:

content. Briefly, 50 μL of the extract (100 μg/mL) was mixed with 150 μL of 95% ethanol,     238

10 μL of 10% (v/v) AlCl3 solution, and 10 μL of 1 M NaOAc solution. Subsequently, the      239

mixture was incubated in the dark for 40 min and the absorbance was measured at 415 nm.     240

  1. Aluminium chloride is a solid substance, so the mode in which the concentration of a AlCl3 solution is presented is wrong, because it is not measured a volume of AlCl3. I suggest to change with “10 μL of 10% (m/v) AlCl3 solution because the correct way to prepare this solution is to measure 10 grams of Al2Cl3 and to dissolve and dilute at 100 mL with water, and in this case the concentration is expressed as mass / weight percent.

Response: The phase “10% (v/v) AlCH3 solution” was incorrect. We have revised it to “10% (w/v) AlCH3 solution” (Revised page 9, line 251).

  1. What is NaOAc solution? From my experience in liquid chromatography I suppose that is sodium acetate, but I am not sure. So, I suggest to write the chemical formula or the name of the substance in bracket.

Response: Thank you for the suggestion. We found some mistakes in the methodology of paragraphs “4.3. Total phenolic contents” and “4.4. Total flavonoid contents”. NaOAc was wrong and the correct was NaNO2.

5) In paragraph 4.11. Phytochemical profiling using LC-MS

at rows 296 - 297 the authors write:

temperature at 40°C using a gradient of (A) 0.1% formic acid and (B) 80% acetonitrile con-    296

taining 0.1% formic acid. The following gradient with a flow rate of 0.2 mL/min was used:    297

I understand the composition of the used mobile phase, I suppose that 80% acetonitrile means a mixture of acetonitrile / water in 80 / 20 ratio. If I am right, I think that is better to write: “using a gradient of (A) 0.1% formic acid and (B) acetonitrile / water (80/20) containing 0.1% formic acid.”

Response: Thank you for the suggestion. We revised this sentence to “using a gradient of (A) 0.1% formic acid and (B) acetonitrile / water (80/20) containing 0.1% formic acid” (Revised page 10, lines 312-313).

6) In paragraph 4.11. Phytochemical profiling using LC-MS

at rows 298 - 299 the authors write:

0−100% B (0−45 min), 100% B (45−50 min), and 0% B (50−60 min). Compounds were puta-   296

tively identified by matching the m/z and fragmentation patterns in mass spectral data         297

I suggest to write “Compounds were calitatively identified by matching the m/z and fragmentation patterns in mass spectral data-bases”

or, simply:

“Compounds were identified by matching the m/z and fragmentation patterns in mass spectral data-bases”

Response: Thank you for the suggestion. We revised this sentence to the following sentences to clarify the data accuracy, as follows:

Revised page 11, lines 315-317

“Compounds were putatively identified by matching the experimental m/z values to the library of theoretical calculated m/z values in databases,”

Reviewer 2 Report

Glochidion Littorale leaf extract was chosen from 20 Thailand plants leaf extracts due to high radical-scavenging activity. Neuroprotective effects of the extracts in Caenorhabditis Elegans were demonstrated. Possible mode of action through DAF-16 activation was suggested. The manuscript is generally well-written, interesting and can be accepted after following issues be addressed.

  1. Abstract. "Phytochemical analysis via LC-MS identified several active compounds..." It should be specified what activity is meant.
  2. Results. The methods used for extracts preparation and screening should be briefly described here.
  3. Table 1. What was the concentration of the extracts? Positive control data for DPPH radical-scavenging activity are missed.
  4. Fig 2. Should be "-" below white column.
  5. Discussion. Some details concerning known Glochidion Littorale extracts activity should be provided.
  6. Oxidative stress assays. H2O2 may be simply neutralized by the extract thus suppressing any biological effect of H2O2. This possibility should be also discussed.

Author Response

We appreciate the time and effort invested by the reviewers and the editor. According to reviewer’s comments, we have carefully revised the manuscript and now believe its quality to be substantially improved. Point-by-point responses to each comment can be found below.

Comments

Glochidion Littorale leaf extract was chosen from 20 Thailand plants leaf extracts due to high radical-scavenging activity. Neuroprotective effects of the extracts in Caenorhabditis Elegans were demonstrated. Possible mode of action through DAF-16 activation was suggested. The manuscript is generally well-written, interesting and can be accepted after following issues be addressed.

  1. Abstract. "Phytochemical analysis via LC-MS identified several active compounds..." It should be specified what activity is meant.

Response: Thank you for valuable suggestion. We have revised “Phytochemical analysis via LC-MS identified several active compounds...” to “The characterization of this extract by LC-MS identified several phenolic compounds” (Revised page 1, lines 29-30).

  1. Results. The methods used for extracts preparation and screening should be briefly described here.

Response: We revised them to the following sentences to describe the methods used for extracts preparation and screening, as follows:

Revised page 2, lines 77-79

“Crude extracts of edible leaves from plants cultivated in Thailand were prepared by ultrasonication. The leaf extracts of 20 plants were screened for their phenolic and flavonoid contents, and antioxidant activity by DPPH radical-scavenging assay.”

  1. Table 1. What was the concentration of the extracts? Positive control data for DPPH radical-scavenging activity are missed.

Response: Since there was no information about the sample concentration in DPPH assays, we added the following sentences to clarify the concentration of the extracts (Revised page 9, lines 259-261).

“Briefly, 100 μL of the extract (1 mg/mL) were mixed with 100 μL of DPPH solution. After 30 min, the absorbance was measured at 517 nm using a microplate reader.”

Thank you for the suggestion concerning positive control data. Since we focused on the comparison among tested samples, the positive control data was omitted in this study.

  1. Fig 2. Should be "-" below white column.

Response: As suggested, we have revised the symbol to “-” in the column of Figure 2.

  1. Discussion. Some details concerning known Glochidion Littorale extracts activity should be provided.

Response: Thank you for pointing this out. As mentioned in the Discussion section (Revised page 8, lines 178-179), there are few studies concerning the functional properties and constituents of Glochidion Littorale.

  1. Oxidative stress assays. H2O2 may be simply neutralized by the extract thus suppressing any biological effect of H2O2. This possibility should be also discussed.

Response: Thank you for valuable suggestion. Actually, the mechanism underlying suppressing effect of oxidative damage is also involved in radical scavenging activity. Therefore, we have revised the sentence to discuss the possibility whether H2O2 was simply neutralized by the extract, as follows:

Revised page 8, lines 186-189

“This result suggests that the antioxidant effect of GLE in reducing oxidative stress in nematodes is possibly involved in not only radical scavenging activity but also the regulation of DAF-16 transcription factor.”